# Genetic Background of Congenital Erythrocytosis

**DOI:** 10.3390/genes12081151

**Published:** 2021-07-28

**Authors:** Mary Frances McMullin

**Affiliations:** Centre for Medical Education, Queen’s University, Belfast BT9 7AB, UK; m.mcmullin@qub.ac.uk

**Keywords:** congenital, familial, erythrocytosis, erythropoietin, oxygen sensing pathway

## Abstract

True erythrocytosis is present when the red cell mass is greater than 125% of predicted sex and body mass, which is reflected by elevated hemoglobin and hematocrit. Erythrocytosis can be primary or secondary and congenital or acquired. Congenital defects are often found in those diagnosed at a young age and with a family history of erythrocytosis. Primary congenital defects mainly include mutations in the *Erythropoietin receptor* gene but *SH2B3* has also been implicated. Secondary congenital erythrocytosis can arise through a variety of genetic mechanisms, including mutations in the genes in the oxygen sensing pathway, with high oxygen affinity hemoglobin variants and mutations in other genes such as *BPMG*, where ultimately the production of erythropoietin is increased, resulting in erythrocytosis. Recently, mutations in *PIEZ01* have been associated with erythrocytosis. In many cases, a genetic variant cannot be identified, leaving a group of patients with the label idiopathic erythrocytosis who should be the subject of future investigations. The clinical course in congenital erythrocytosis is hard to evaluate as these are rare cases. However, some of these patients may well present at a young age and with sometimes catastrophic thromboembolic events. There is little evidence to guide the management of congenital erythrocytosis but the use of venesection and low dose aspirin should be considered.

## 1. Introduction

The normal range for hemoglobin (Hb) and hematocrit (Hct) is defined for men and woman as the mean plus or minus two standard deviations and therefore encompasses 95% of the population. Those falling outside this normal range are considered abnormal. In the case of an Hb or Hct above the normal range for sex, it is considered that there is elevated Hb. However, in order to prove that the Hb is elevated above normal, it is necessary to measure the red cell mass. By definition, red cell mass is considered to be increased if it is predicted to be greater than 125% for sex and body mass [1] and this is referred to as a true erythrocytosis.

Unfortunately, it is becoming increasingly difficult, for operational reasons, to determine red cell mass. Increased Hb or Hct does not always correlate with increased red cell mass, as has been shown in studies by Johansson et al. in subjects where both results were available [2]. However, when Hct is greater than 0.60 (60%) in men and 0.56 (56%) in females, this has been found to always correlate with increased red cell mass. At lower levels in the absence of a red cell mass measurement, it is often assumed that there is erythrocytosis.

Erythrocytosis can be primary or secondary and congenital or acquired (Figure 1). A primary erythrocytosis is where an intrinsic defect in the stem cell population leads to an increase in red cell production. The most common acquired primary erythrocytosis is polycythemia vera and in these cases the subject’s erythropoietin (EPO) level will be below the normal range because the defect is intrinsic to the stem cells. Secondary erythrocytosis is present when erythropoietin (EPO) is being produced and driving red cell production. Secondary acquired erythrocytosis has many possible causes where EPO from some source drives red cell production and EPO levels are elevated or inappropriately normal (an EPO level of normal is inappropriate for Hb above the normal range).

Congenital erythrocytosis is likely to occur in someone presenting at a young age and/or with a family history of erythrocytosis. There are a number of genetic lesions which have been described to cause both primary and secondary erythrocytosis and this review will provide an update on the current known lesions.

After investigation, there remains a group with erythrocytosis in whom no cause is identified and this group are referred to as idiopathic erythrocytosis. They remain a group in whom further investigation is warranted.

## 2. Primary Congenital Erythrocytosis

The genetic lesions in congenital erythrocytosis or congenital and familial erythrocytosis, as it can be referred to, are coded as erythrocytosis (ECYT). Primary genetic disorders in the *erythropoietin receptor gene* (*EPOR*) are ECYT1 and ECYT 2–8 are secondary disorders, with genetic lesions leading to secondary erythrocytosis.

### 2.1. EPOR

A primary congenital erythrocytosis arises when there is a germ-line intrinsic defect in the bone marrow compartment, leading to increased red cell production without the need for an extrinsic EPO drive (therefore, the EPO level will be below the normal range). The most common reason for this is a mutation in the *Erythropoietin Receptor* (*EPOR*) gene. The first described reason for congenital erythrocytosis was in fact a mutation in the *EPOR* gene [3], also known as *ECYT1*. This mutation was discovered in a Finnish, multiple gold medal winning, Olympic skier Eero Mantyranta who had elevated Hb. Multiple family members were affected and the causative genetic mutation was discovered in the *EPOR* with a G to A transition at nucleotide 6002, resulting in a stop codon. This mutation was found in all affected family members. The same mutation was subsequently discovered, arising independently in a young English man [4]. This stop codon results in a 70 amino acid truncated erythropoietin receptor which has lost the docking site for the protein SHP1, the protein which terminates signaling. Hence, the receptor lacks the negative regulatory domain. The receptor is then continuously signaling, resulting in increased red cell production in the absence of the EPO ligand.

Subsequently a number of further mutations have been reported in the *EPOR* gene, which would be predicted to act in a similar fashion, leading to a truncated erythropoietin receptor. These have been delineated in previous publications [5].

Missense mutations have also been described in the *EPOR*. It is not always clear if these contribute to erythrocytosis. However, extensive functional studies show EPO hypersensitivity with a least one new mutant, p.Gln434Profs*11. This germline heterozygous mutation in a patient with erythrocytosis acts as a gain-of-function mutant [6].

### 2.2. LNK/SH2B3

The lymphocyte adaptor protein (LNK) (also named SH2B3) has a role in regulation of hematopoiesis acting as a negative regulator of cytokine receptor mediated JAK/STAT signaling. *LNK* mutations have been described in patients with idiopathic erythrocytosis and may account for the erythrocytosis. It is not clear in cases if these mutations are germ-line, although *LNK* mutations in other myeloproliferative neoplasms have been found in the germ-line [7]. However, *LNK* mutations may alter LNK downstream of the EPOR resulting in upregulation of the JAK/ STAT pathway and therefore cause primary erythrocytosis.

## 3. Secondary Congenital Erythrocytosis

### 3.1. The Oxygen Sensing Pathway

Eukaryotic cells have mechanisms for the maintenance of oxygen homeostasis. This process involves a number of different proteins, prolyl hydroxlases (PHD), von Hippel-Lindau (VHL) and hypoxia inducible factors (HIFs) in a regulatory pathway. PHD proteins which exist in three isoforms (PHD1, PHD2, and PHD3) have an absolute requirement for oxygen and in normal oxygen conditions hydroxylate HIF-α at the oxygen-dependent domain of HIF-α. Following this, VHL associates with HIF-α, providing a recognition site for the ubiquitin ligase complex. The HIF-α is then degraded by ubiquitination in the proteasome.

However, in hypoxic conditions, prolyl hydroxylation does not occur and PHD cannot associate with VHL and ultimately go down the pathway of ubiquitination. Instead, HIF-α accumulates and associates with HIF-β. HIF-α and HIF-β form a stable HIF complex which translocates to the nucleus and binds to promoters and enhancers of a range of genes. This leads to transcription of a number of genes and protein production. These genes and proteins include those involved in hormone regulation, energy metabolism, angiogenic signaling growth and apoptosis and cell regulation. EPO is among the proteins produced as a response to hypoxia and therefore drives red cell production and increases oxygen carrying capability [8].

Mutation in the genes in the oxygen sensing pathway could lead to abnormal protein production, which would not undergo ubiquitination in states of normal oxygen tension but instead behave as if in a state of hypoxia. Such mutations in each of these genes have been described (Table 1).

### 3.2. VHL

Mutations in the *VHL* gene lead to secondary erythrocytosis. The inheritance is autosomal recessive and this is coded as ECYT2. The first mutation in *VHL* was found to be linked to erythrocytosis in the Chuvash region of Russia where there were a large number of cases of erythrocytosis. All affected cases had a homozygous mutation C598T which results in an Arg200Trp change. In vitro experiments with the mutant protein showed that it had impaired interactions and reduced ubiquitination in normoxic conditions [9].

Clinical phenotypes were studied in this area where there were numerous homozygotes and heterozygotes. Homozygosity was associated with varicose veins, lower blood pressure, vertebral hemangiomas and premature mortality related to cerebrovascular events. It was not associated with hemangioblastomas, renal carcinomas and pheochromocytomas as seen in classical von Hippel-Lindau syndrome [10].

The same mutation was seen in other individuals of Asian and European descent [11] and a large cohort in the Italian island of Ischia [12]. Studies showed that there was likely to have been a single founder arising approximately 50,000 years ago [13]. A number of other *VHL* mutations have identified as being associated with erythrocytosis, including other homozygote lesions and compound heterozygotes (listed in the work of Bento, 2014) [5].

In more recent times, seven families with erythrocytosis and one large family with von Hippel-Lindau disease have been described with mutations in a *VHL* cryptic exon, which produced dysregulation of *VHL* splicing associated with downregulation of VHL protein expression [14].

### 3.3. PHD2/EGLN1

Erythrocytosis associated with a *PHD2* (also known as *EGLN1*) mutation was described across generations in multiple members of a family. A heterozygous C to G change at base 950 resulted in a change at codon 317: P317R. Experimental in vitro work demonstrated that the mutated protein was likely to be the cause of the erythrocytosis [15]. The mutated PHD enzyme has a marked decrease in activity. The phenotype of affected individuals was relatively mild erythrocytosis and EPO levels within the normal range. Nevertheless, the mutation would account for the erythrocytosis seen. This autosomal dominant disorder is coded as ECYT3. A number of other *PHD2* mutations associating with erythrocytosis were subsequently discovered in families with erythrocytosis [5,16,17]. A mouse model of the P317R *PHD2* mutation resulted in erythrocytosis [18].

A 14-year-old female with erythrocytosis was found to be homozygous for a missense mutation in the PHD2 zinc finger (T124C: C42R). The healthy consanguineous parents were both heterozygotes. It is proposed that the zinc finger allows recruitment of PHD2 to pathways to facilitate hydroxylation of HIF-α [19].

The phenotype of these *PHD2* mutated patients varies with some thrombotic events but of note there are reports of paragangliomas and a pheochromocytoma in one patient [20,21]. Loss of heterozygosity in the wild type allele combined with the mutant allele occurs in the tumors and demonstrates the oncogenic potential which can be associated with a *PHD* mutation.

### 3.4. HIF2A/EPAS1

The first gain-of-function mutation in humans in a *HIF* gene was described in *HIF2A* (also known as *EPAS1*) in a family with three generations of erythrocytosis. A heterozygous change in G1609T with a Gly537Trp amino acid change was seen. In vitro studies showed that the mutant protein was more highly expressed and more slowly degraded and resulted in increased EPO signaling [22]. This is coded as autosomal dominant ECYT4. Other mutations associated with *HIF2A* have been described in cases of erythrocytosis and this will continue to be seen when further cases are investigated [5,22]. A mouse model of the G537W (G536W in the mouse) resulted in erythrocytosis in heterozygous mice [23].

The phenotype is described in the individual families and some severe and early thrombotic events are described [22]. Pulmonary hypertension is another phenomenon seen [24]. *HIF2A* mutations have been associated with pheochromocytomas and paragangloimas found in some patients with germline mutations [25,26].

### 3.5. ERYTHROPOIETIN Gene

An extended Norwegian family, who have been investigated, were found to have a gain-of-function variant in the *ERYTHROPOIETIN* (*EPO*) gene itself. This is autosomal dominant and coded as ECYT5. In a pedigree with 10 affected members across four generations with erythrocytosis, a mutation in *EPO* co-segregated with disease. A mutation (c.32delG) (single nucleotide deletion) results in a frameshift in exon 2 of *EPO*. This interrupts translation of the main messenger RNA but initiates excess production of EPO from an alternative promotor in what would normally be a non-coding EPO mRNA. The increased EPO production drives red cell formation [27]. Thus, a mutation in the *EPO* gene results in secondary erythrocytosis.

### 3.6. High Oxygen Affinity Variants

Oxygen is transported to the tissues bound to Hb. Oxygenation and deoxygenation occur at the heme iron binding site and the affinity for oxygen depends on the particular Hb. Some Hbs have a higher than normal affinity for oxygen. The Hb–oxygen affinity is expressed by the Hb–oxygen dissociation curve. A high affinity Hb has a left shifted Hb–oxygen dissociation curve. Oxygen is tightly bound and therefore not released to the tissues. The result is tissue hypoxia, EPO production and erythrocytosis.

Hb Chesapeake was the first reported high oxygen affinity Hb described in 1966 [28]. Over 100 high oxygen affinity Hb variants have been described. Both α and β variants with high oxygen affinity are seen and such high oxygen affinity variants are coded as ECYT7. These variants are inherited in an autosomal dominant manner and are frequently associated with a family history of erythrocytosis. In the past, these variants, as a cause of erythrocytosis, were often missed [29].

### 3.7. Methemoglobinemia

Methhemoglobin, which is formed when ferrous heme is oxidized, impairs oxygen binding and transport. Methemoglobinemia will result in compensatory erythrocytosis. It can occur as an inherited disorder either due to an abnormal M Hb variant or due to an inherited deficiency of a cytochrome reductase. M Hb variants are inherited in an autosomal dominant fashion and can be α, β, γ variants. Cytochrome reductase deficiency type I is confined to erythrocytes and is a benign disorder with cyanosis, whereas type II deficiency affects all cells and presents with a severe neurological disorder and cyanosis [30].

### 3.8. Bisphosphoglycerate Mutase Deficiency

Bisphosphoglycerate mutase (BPMG) catalyzes the synthesis of 2,3-BPG from 1,3-BPG in red cells. 2,3-BPG binds deoxy Hb and reduces its affinity for oxygen. Deficiency of BPMG leads to reduced 2,3-BPG and a shift of the oxygen dissociation curve of Hb to the left. This means less oxygen is available to tissues and therefore EPO levels increase to drive red cell production and increase oxygen supply. In those with deficiency of BPMG, mutations in the *BPMG* gene have been described as resulting in defective protein production and these mutations account for the erythrocytosis in these patients [31]. These mutations are listed as ECYT8. A case with uniparental disomy with a novel *BPGM* mutation leading to a homozygous state has been described in a case of congenital erythrocytosis [32].

### 3.9. PIEZO1

The *PIEZO1* (Piezo type mechanosensitive ion channel component 1) gene encodes for the PIEZO1 protein, which is a mechanosensitive ion channel translating a mechanical stimulus into calcium influx. Gain-of-function mutations have been described in hereditary xerocytosis. In the description of a large series with this disorder some patients were noted to have Hb levels at the upper limit of normal or above normal and signs of hemolysis [33]. A recent study used a cohort of 110 patients with idiopathic erythrocytosis and a systematic search revealed that 4% had pathogenic mutations in the *PIEZO1* gene. These patients had features of hemolysis as well as increased Hb. Of note, the venous P_50_ was reduced, indicating an increased oxygen affinity. This would lead to tissue hypoxia and compensatory EPO driven erythrocytosis [34].

### 3.10. SLC30A10 Mutations with Hypermanganesemia

In the syndrome of hepatic cirrhosis, dystonia, polycythemia and hypermanganeseemia, mutations in the *SLC30A10* gene, a manganese transporter in humans, have been identified in a number of families. These mutations lead to increased manganese levels and manganese toxicity causes the clinical features. Manganese is suggested to induce *EPO* gene expression, leading to the increased EPO levels and erythocytosis described with the syndrome. Identification of this syndrome is of importance as chelation of manganese should be attempted [35].

## 4. Management of Congenital Erythrocytosis

Congenital erythrocytosis cases and families are very rare and it is therefore difficult to gain a perspective on the phenotypes and outcomes. Retrospective studies in patient cohorts from Chuvashia, Russia, who are homozygous for the *VHL* mutation R200W, have been found to have a number of clinical phenomena including varicose veins and thrombotic events.

With this and other mutations of the oxygen sensing pathway, reports suggest that serious thromboembolic events occur at unusual sites in young patients. Mortality and morbidity associated with thromboembolic events are increased in affected individuals [36]. There are indications that in the long term other medical complications such as pulmonary hypertension may emerge, and in some cases, neuroendocrine tumors.

Physiological studies on *VHL* and *HIF2* mutated patients showed that, in Chuvash, polycythemia was highly characteristic of acclimatization to the hypoxia of high altitude [37] and *HIF2* mutations were associated with pulmonary hypertension [38].

There are few management options in this very rare group of patients with congenital erythrocytosis. Venesection would be the logical therapeutic intervention to reduce the Hct and, by implication, the viscosity. However, this may not be of benefit as the altered physiology due to the mutations may require a higher Hb for satisfactory oxygen carriage to tissues [37]. Venesection is only suggested if a decrease in Hb is associated with improvement in symptoms [39].

Low dose aspirin is proven to reduce the incidence of thromboembolic events in acquired myeloproliferative neoplasms [40]. It seems logical to give this to those with congenital erythrocytosis in whom there is no contraindication to aspirin as it may be of benefit but evidence for this benefit is very difficult to generate in such rare disorders.

The VHL protein acts upstream of the JAK/STAT pathway [41] and therefore the JAK inhibitor ruxolitinib may have a role in downregulation of the pathways in patients with mutant VHL protein. There have been some cases reported of Chuvash polycythemia treated with ruxolitinib who had useful improvements in symptoms and blood parameters [42]. This may be a potential therapeutic option in such Chuvash patients with problematic diseases.

There may be more specific therapeutic options to inhibit mutant proteins in the oxygen sensing pathway in the future. Small molecule inhibitors of PHD2 are becoming available for the treatment of anemia [43]. In clear cell renal carcinoma, HIF2a accumulates and small molecule inhibitors of HIF-2a, such as belzutifan, are in clinical trials [44]. As some of these compounds become available for clinical use, they may have a role in inhibiting mutant proteins in specific defects.

## 5. Idiopathic Erythrocytosis

There remain, in clinical practice, a number of patients in whom no cause for their erythrocytosis can be elucidated. Over time, as more lesions have been identified, the number of such patients has decreased. They should be extensively investigated and as new techniques become available and genetic lesions are discovered, more insight is gained into individual patients [45]. Other genetic factors may be of importance. *HFE* mutations are frequently observed in idiopathic erythrocytosis and may in some way facilitate increased red cell mass [46,47]. Nevertheless, the clinical problem remains as to how to manage patients with idiopathic erythrocytosis in the clinic. The evidence for management does not exist but guidelines suggest aspirin if there is no contraindication and venesection in selected cases with the HCT target tailored to the thrombotic history and risk factors [39].

## 6. Conclusions

An increasing number of genetic lesions have been found over time that can account for erythrocytosis. Patients with proven erythrocytosis, particularly in the young and in those with a family history of it, should be extensively investigated for genetic causes and re-investigated whenever further discoveries are made. Over time, perhaps the cause will be found even in those currently with idiopathic erythrocytosis.

Best management is difficult to determine given the rarity of these cases. There remains a need for further long-term follow-up in order to obtain better outcome data.

## Figures and Tables

**Figure 1 genes-12-01151-f001:**
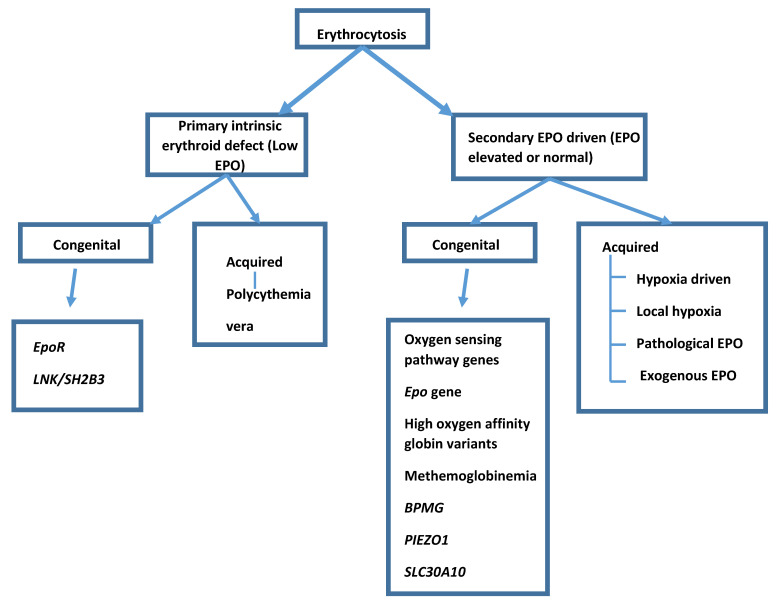
Classification of erythrocytosis.

**Table 1 genes-12-01151-t001:** Gene associated with erythrocytosis.

Primary		
	*Erythropoietin receptor* *LNK/SH2B3*	
Secondary		
	Oxygen sensing pathway genes	
		*VHL* *PHD2/EGLN1* *HIF2A/EPAS1*
	*Erythropoietin gene*	
	α and β globin genes (High oxygen affinity Hbs) Methemoglobinemia (Abnormal M Hb or cytochrome reductase deficiency) *BPMG* *PIEZ01* *SLC30A10*	

## Data Availability

Not applicable.

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
