# Peer review of "Genetic Background of Congenital Erythrocytosis"

_genes, 2021, doi:10.3390/genes12081151_

Round 1

Reviewer 1 Report

A comprehensive, excellent  and well-written paper by world's  leading expert in the field. This Reviewer has no further comments other than the paper is warmly recommended for publication.

Author Response

Thank you for your kind comments and acceptance of paper. Some improvements have been made in line with reviewer 2

Reviewer 2 Report

In this review paper, the author is providing comprehensive information about congenital erythrocytosis with recent findings. While this reviewer has read this review with interest and recognizes the importance of this topic, the current manuscript should be improved, particularly in terms of the presentation of figures and tables.

The current figure 1 seems to be too simple to deliver essential concepts to readers. What distinguishes between primary and secondary erythrocytosis?

What kind of genetic lesions are involved in primary and secondary congenital erythrocytosis? These should be included in the figure with an appropriate explanation and/or legend.

Revision on Tables is also required. Although the title of Table 1 is “Gene mutations causing erythrocytosis”, current Tabe1 only shows a list of genes. As no information about mutations in those genes is provided, it is completely obscure how those mutations affect erythropoiesis. Such information should be included concisely with appropriate references. Table 2 is also not appealing. Ideally, it should include proper application and possible benefits/drawbacks of those therapies with references.

Minor

  1. Line 19

“to guide the management of management of a congenital erythrocytosis”

----- “to guide the management of a congenital erythrocytosis”

  1. Line 35-36

when the Hct is greater than 0.60 in men and 0.56 in females

----- 0.60 and 0.56 mean 60% and 56%?

  1. Line 74-75

“for the protein SHP1 the protein which terminates signalling”

----- “for the protein SHP1 which terminates signalling”

  1. Line264

ruxulitinib ----- ruxolitinib

Author Response

please note I am uploading new figure 1 separately as I cannot get it to paste in to the manuscript

Round 2

Reviewer 2 Report

The author has amended the manuscript according to my comments. I do not have further criticisms on this paper.